# The metabolic pathways of carbon assimilation and polyhydroxyalkanoate production by *Rhodospirillum rubrum* in response to different atmospheric fermentation

**Manyu Tang**[1]**, Xin Zhen**[1]**, Guoqiang Zhao**[1]**, Shuang Wu**[1,2]**, Wei Hua**[1,2]**, Jingwen Qiang**[1]**, Cheng Yanling**[1,2]*****, Wanqing Wang**[1,2]*

**1** Biochemical Engineering College, Beijing Union University, Beijing, China, **2** Beijing Key Laboratory of Biomass Waste Resource Utilization, Beijing, China

* shtwanqing@buu.edu.cn (WW); cheng1012cn@aliyun.com (CY)

**Data Availability Statement:** All relevant data are within the manuscript and its Supporting Information files.

## Abstract

The purple nonsulfur bacteria, *Rhodospirillum rubrum*, is recognized as a potential strain for PHAs bioindustrial processes since they can assimilate a broad range of carbon sources, such as syngas, to allow reduction of the production costs. In this study, we comparatively analyzed the biomass and PHA formation behaviors of *R. rubrum* under 100% CO and 50% CO gas atmosphere and found that pure CO promoted the PHA synthesis (PHA content up to 23.3% of the CDW). Hydrogen addition facilitated the uptake and utilization rates of CO and elevated 3-HV monomers content (molar proportion of 3-HV up to 9.2% in the presence of 50% $H_2$). To elucidate the genetic events culminating in the CO assimilation process, we performed whole transcriptome analysis of *R. rubrum* grown under 100% CO or 50% CO using RNA sequencing. Transcriptomic analysis indicated different $CO_2$ assimilation strategy was triggered by the presence of $H_2$, where the CBB played a minor role. An increase in BCAA biosynthesis related gene abundance was observed under 50% CO condition. Furthermore, we detected the α-ketoglutarate (αKG) synthase, converting fumarate to αKG linked to the αKG-derived amino acids synthesis, and series of threonine-dependent isoleucine synthesis enzymes were significantly induced. Collectively, our results suggested that those amino acid synthesis pathways represented a key way for carbon assimilation and redox potential maintenance by *R. rubrum* growth under syngas condition, which could partly replace the PHA production and affect its monomer composition in copolymers. Finally, a fed-batch fermentation of the *R. rubrum* in a 3-l bioreactor was carried out and proved $H_2$ addition indeed increased the PHA accumulation rate, yielding 20% ww$^{-1}$ PHA production within six days.

**Funding:** This work was supported by grants from the National Natural Science Foundation of China (31600208); The Project of Cultivation for young top-motch Talent of Beijing Municipal Institutions (BPHR202203210).

**Competing interests:** The authors have declared that no competing interests exist.

## Introduction

Valorization and reuse of wastes through bioconversion into value-added products plays an extremely important role in releasing the current energy crisis and environmental pollution [1, 2]. Syngas, which is a blend of CO and $H_2$, is widely formed during thermochemical conversion of various wastes and is considered an energy vector for a sustainable energy future. Syngas could be transformed into value-added products including ethanol, butanol, acetic acid or butyric acid by carbon-fixing microorganisms, in a process known as syngas fermentation [3, 4]. So, syngas fermentation presents a highly attractive potential for biofuel, fine chemicals and biopolymer production nowadays [5, 6].

Poly(3-hydroxyalkanoates) (PHAs) considered "green plastics", are one of the potential products that can be synthesized by syngas fermenting bacteria [7, 8]. *Rhodospirillum rubrum* (*R. rubrum*), a purple non-sulfur bacteria (PNSB) alpha-proteobacterium, can use syngas as carbon and energy source for growth and for PHAs accumulation under anaerobic [9, 10]. *R. rubrum* is equipped with CO monoxide dehydrogenase (CODH) and hydrogenase ($H_2$ase) activity to fix syngas [11, 12]. Applying the water-gas shift reaction, it oxidizes CO with $H_2O$ into $CO_2$ and $H_2$. Then, the produced H is later used to assimilate carbon molecules from CO and $CO_2$ into the biomass. $CO_2$ can then be fixed by various carboxylases into the ethylmalonyl-CoA pathway, the Calvin-Benson-Bassham cycle or the reductive tricarboxylic acid cycle to produce PHA [10, 13, 14]. During syngas fermentation, *R. rubrum* mainly synthesizes the short-chain-length PHAs (PHA$_{SCL}$; 3 to 5 carbon atoms), where 3-hydroxybutyrate (3HB) is a dominant monomer of the product. The product in RRNCO medium with syngas was reported to contain 86% of 3HB and 14% 3HV (3-hydroxyvalerate) of total PHA [10]. The presence of 3HV occurrence in the polymer enhanced physicochemical properties (e.g., higher elasticity, flexibility, etc.) [15]. Furthermore, the components of carbon in the precursor substrate greatly influences the amount of carbon in the monomers in the PHA chain.

Currently, the main challenges of the industrial-scale syngas to PHAs conversion were the low syngas conversion efficiency and the slower PHA synthesis rate of the associated microbes [16]. It was reported that CO conversion efficiency was around 50% when artificial syngas was used as the carbon source [10, 17]. To overcome this limitation, more information about the assimilation metabolism of syngas is needed. Previous research showed that the ribulose 1,5 biphosphate carboxylase/oxygenase (Rubisco) was mainly involved in $CO_2$ fixation during photoautotrophic growth [18, 19]. In addition to the Calvin–Benson–Bassham (CBB) cycle, ethylmalonyl-CoA pathway (EMC) is now well accepted as an alternative pathway for the $CO_2$-fixing to achieve electron balance during phototrophic growth on acetate [20, 21]. Moreover, acetate has been shown to be a good co-substrate for *R. rubrum* growth in the process of syngas fermentation. It is mainly assimilated through the EMC pathway to provide the carbon skeleton for PHAs and biomass synthesis [21, 22]. In addition to microbial metabolism, the composition of syngas (commonly referred to in terms of $H_2$/CO ratio), which varies depending on its origin or substrate, is another non-ignored factor in the actual fermentation process. Moreover, previous studies have demonstrated that different syngas concentrations have a significant impact on the growth of *R. rubrum* [14]. The experiments with different initial CO concentrations have revealed that a $p_{CO}$ of 0.60 atm was optimal condition for *R. rubrum* cultivation in darkness and its maximum polyhydroxybutyrate (PHB)production reached 26% CDW [23]. Additionally, research on *R. rubrum* grown in different partial pressures of CO has also shown that the maximal biomass and PHA yield (8.3% CDW) were obtained at ppCO 0.6 bar under light condition [24].

The main objective of this research was to study the assimilation of syngas in *R. rubrum* under different atmosphere conditions directed towards CO utilization rates, PHAs

production, as well as, to demonstrate $CO_2$ assimilation strategy. In order to reveal the assimilation of CO into cellular central metabolites biomass, a comparative transcriptomic analysis of 100% CO and 50% CO fermentation was carried out. We highlighted two different assimilation pathways for CO photo-assimilation. Under pure CO condition, Rubisco was actively incorporating $CO_2$ from gas substrate into biomass and the bioproduction of PHA acted as an electron sink in the balance of reduction equivalents during photo-heterotrophic growth. Instead, the branched-chain amino acids (BCAA) biosynthesis as an unexpected new $CO_2$ assimilation pathway was significantly induced under 50% CO (50% $H_2$) condition and partly substituted for PHA synthesis contributing to electron balance. Finally, a fed-batch fermentation revealed a decrease in cellular PHA content but a prominent increase in PHA synthesis rate with 25% $H_2$ added in initial gas. And its whole fermentation period could be cut down by half to six days, yielding the final PHA yield around 20% w $w^{-1}$.

## Materials and methods

### Bacterial strain, growth conditions, and media

In this study, *R. rubrum* (ATCC 11170) was used for all fermentations. A two-step culture strategy was preferred to accumulate PHA. Frist, an enriched broth medium (Trypticase soy broth medium contains per liter of distilled water: 17.0 g tryptone, 3.0 g soytone, 2.5 g glucose, 5.0 g sodium chloride and 2.5 g dipotassium phosphate) was used for rapid microbial heterotrophic growth (at 28˚C and 180 rpm aerobically dark growth for 3–4 days) until the culture reaches the stationary phase ($OD_{600}$ 1.2–1.5).

Then a modified RRNCO medium as described previously supplemented with 1.38 g $l^{-1}$ (~16.8 mM) sodium acetate was adopted for subsequent microbial anaerobic photoheterotrophic cultivation to accumulate PHA. The composition of RRNCO medium was as follows (per liter of distilled water): 5.23 g $NaH_2PO_4·2H_2O$, 11.55 g $Na_2 HPO_4·12H_2O$, 0.31g $K_2SO_4$, 0.04 g NaOH, 0.1 g $NH_4Cl$, 0.79 g $MgSO_4·7H_2O$, 0.061 g $CaCl_2·2 H_2O$, and 1ml trace metal solution (0.048 g $CuSO_4·5H_2O$, 0.24 g $ZnSO_4·7 H_2O$, 0.24 g $MnSO_4$ and 1.5 g $FeSO_4·7H_2O$ per 100 ml), maintained pH around 7.0.

For growth-induction transition experiments, the cultures of broth medium were centrifuged at 5000 rpm for 5min, and the pellet were washed once with saline solution (NaCl 0.9%). Then the cells were resuspended with RRNCO medium and fructose (10 g $l^{-1}$) or CO were separately added as the fermentation carbon source for cellular PHA synthesis. Fructose fermentation cultures were shaken at 28˚C and 180 rpm in dark, but no strictly anaerobic growth. Gas fermentation experiments were done in bottles of 100 ml containing 20 ml of RRNCO medium (supplied with 10 mM acetate). Prior to adding gas, the closed degasified serum vials were re-filling with $N_2$ followed by subjected to 3 min vacuum-purge. After a third vacuum cycle, the atmosphere was saturated with various fermentation gas (100% CO, 75% CO, 50% CO or 20% CO) to 1 atm of pressure. Incubation followed in a white light shaker (light intensity ~30 μE) at 28˚C and 180 rpm for continuous fermentation.

### Fed-batch experiments in bioreactor

The experiment with different gas components were carried out in a 3-l glass-made bioreactor. The schematic diagram of the experimental setup is shown in Fig 8A. Cells for fed-batch experiments were prepared using the same operational conditions for pre-cultures of *R. rubrum* in broth medium (i.e., temperature, shaker speed and initial biomass concentration). Then the 3-l stationary phase grown pre-culture were centrifuged and resuspended with 0.6-l RRNCO medium in bioreactor for gas fermentation. The vessel was sterilized by autoclaving at 121˚C for 20 min before inoculation. Similarly, the closed glass bioreactor was subjected to 3 min

vacuum-purge before adding gas. The initial gas supply controlled by two mass flow controllers with 0.1vvm (100 ml min$^{-1}$) gas flow. For pure CO fermentation, 2.4 liters of CO was injected into the bioreactor at once. For the simulated syngas fermentation, 1.8 liters of CO and 0.6 liters of $H_2$ were respectively injected into bioreactor. The fermentational conditions were indicated as follows: 28˚C of temperature, 250 rpm of stirrer speed and light intensity maintained around 30 uE by white led lamps. For the acetate supplement experiment, sodium acetate, previously degassed with $N_2$ and filtered to sterility (Syringe Filter 0.22 μm) was added to reach a final concentration of 10 mM. Notably, at the end of syngas fermentation, residual gas was composed of 35% CO, 48% $H_2$ and 17% $CO_2$. Therefore, a preliminary outlet gas reflux experiment was designed. 1.2 liters of residual gas ($CO_2$ removed by zeolite molecular sieves) was returned by a gas circulation pump and meanwhile 1.2 liters of CO was injected into the bioreactor to conduct the next round syngas fermentation.

## Gas analysis

The gases hydrogen, carbon monoxide and carbon dioxide were analyzed by using a gas chromatography (GC-2014 Shimadzu, Japan) on a stainless-steel packed column TDX-01(3.0 m × 3.00 mm) with a thermal conductivity detector (TCD). The parameters were set as 200˚C at the inlet, 100˚C at the column, 200˚C at the detector, 35 ml/min at the carrier gas, 50 mA at the flow rate, and 40 min at the data collection time. The samples were collected from the top space of the culture at different times, using a 1 ml air-tight needle to extract headspace gas from the anaerobic fermenter for gas phase analysis, and using a 1 ml air-tight needle to extract different volumes of gas before measurement, manual sampling analysis, the establishment of standard curve. The standard curve was drawn, and the gas concentration was calculated by using the gas chromatography data analysis software (Lab solution).

## PHA extraction and quantification

PHA extraction and quantification assay were performed as our previously described [23]. Briefly, 10 to 20 mg of freeze-dried cell matter was methanolized, adding 2 ml methanol (containing 15% sulfuric acid and 0.5 mg ml$^{-1}$ of 3-methylbenzoic acid (internal standard)) and 2 ml chloroform, at 100˚C, esterification for 4 h. After cooling, the same volume of demineralized water was added. The organic phase was filtered with 0.22 μm organic phase filter membrane, injected with automatic sampler, and analyzed by gas chromatography (GC) analysis. GC-2014 (Shimadzu, Japan), RTX-5(30m × 250μm × 0.25 μm) and FID (flame ionization detector) were used in this experiment. The conditions of GC analysis were as follows: 1 μl of sample, 280˚C of inlet and 280˚C of detector, 1ml min$^{-1}$ of flow rate of nitrogen, 1:1 split ratio. Samples with known concentrations of commercial poly(3-hydroxybutyrate-co-3-hydroxyvalerate) (PHBV) (8 mol% PHV, Aladdin) were used to calculate the process efficiency. The retention time for each methylate monomer obtained in this work was 7.1 min (C4), 8.2 min (C5) and 10.2 min (3-methylbenzoic acid, internal standard). The amount of the monomeric composition of PHBV was verified with the standard curves generated by using 3-HB (from 0.5 to 5 mg, J&K$^®$ Scientific) and 3-HV (from 0.05 to 0.5 mg, TRC).

The GC analysis was validated using crotonic acid and 2-pentenonic acid method by HPLC as Godoy et al. reported. Briefly, the biomass of 800 μl of culture was resuspended in a solution of NaOH (1M) and incubated at 94˚C for 1 h. After neutralized by the same volume of HCl (1N). The neutralized samples were centrifuged at 2600 rcf for 10 min and the supernatants were filtered through 0.45 μm pore filters. The identification and quantification of crotonic acid and 2-pentenonic acid were determined by HPLC (20A, SHIMADZU) with an Aminex HPX-87H column (Biorad, Hercules, CA, USA). The mobile phase was $H_2SO_4$ (5 mM), and

the flow rate used was of 0.6 ml·min$^{-1}$. The detection was performed by a refractive index detector. The compounds were identified by their retention time, and the concentration was estimated with standard curves prepared with HPLC-grade compounds: crotonic acid (TCI, Shanghai) and 2-pentenoic acid (TCI, Shanghai).

## Acetate content analysis

The supernatant of culture broth was filtered through a 0.22 μm syringe filter and stored at -20˚C. A high-performance liquid chromatography (HPLC) equipped with an ion exchange column (Aminex HPX-87H, BioRad) and a refractive index detector (RI-150, Thermo Spectra System, USA) was used for acetate concentration analysis. A mobile phase of 5 mM $H_2SO_4$ at a 0.5 ml min$^{-1}$ flow rate was used.

## Microarray analysis

The RNA samples were collected from cells harvested: the enriched broth medium growth for 3 d, fructose-induced fermentation for 108 h, pure CO-induced fermentation for 8 d and 50% CO-induced fermentation for 8 d. In each condition, 4 ml of the culture was collected, and total RNA was extracted using the RNAprep Pure Cell/Bacteria kit (Tiangen). After rRNA removal, total RNA was reverse transcribed to obtain cDNA using PrimeScript Double-Strand cDNA Synthesis Kit (Tiangen) for RNA-Seq.

An Illumina TruSeq kit (Illumina) was used to prepare RNA-seq libraries in three biological replicates and the Illumina HiSeq4000 platform was used for sequencing. Gene expression levels were compared and normalized using the R statistical package edgeR. DEGs were identified as significantly expressed when their ration log2 (100% CO or 50% CO/control) shown > 1 or < − 1, respectively, and their FDR adjusted P value (Q value) was < 0.05.

## Real-time qRT-PCR assays

Total RNA was extracted using a RNAprep Pure Cell/Bacteria kit (Tiangen), and the first-strand cDNA was synthesized by reverse transcriptase (Invitrogen). Real-time PCR was performed using the FastFire qPCR PreMix (SYBR Green) (Tiangen) following the manufacturer's instructions. To determine a suitable reference gene, thirteen candidate genes were assessed under different growth conditions to evaluate their stability. The evaluated genes were CO-sensing transcription activator (*Rru_A1431*), ribulose bisphosphate carboxylase (*Rru_A2400*), crotonyl-CoA carboxylase/ reductase (*Rru_A3063*), pyruvate-flavodoxin oxido-reductase (*Rru_A2398*), propionyl-CoA carboxylase (*Rru_A1943*), 2-oxoglutarate synthase (*Rru_A2721*), PHA synthase (*Rru_A0275*), PHA polymerase homologous gene (*Rru_A1816*), β ketothiolase (*Rru_A0274*), acetoacetyl-CoA reductase (*Rru_A0273*), carbon-monoxide dehydrogenase (*Rru_A1427*) and pyruvate carboxylase (*Rru_A2317*). The expression levels were normalized to the expression of a 16S rRNA gene (*Rru_A0064*). Primers are listed in Supplemental Data S4 Appendix.

## Result and discussion

### PHA production of *R. rubrum* for using CO as carbon substrate

*R. rubrum* is able to utilize CO as carbon and energy sources by the water-gas shift reaction and various carbon-fixing metabolisms to produce PHA. The biomass and PHA formation behaviors of *R. rubrum* under different carbon substrate were investigated (Fig 1). The bacteria were firstly cultured in growth medium 3 days to achieve a high cell density, then transferred to the induced medium containing fructose or CO as the sole source of carbon and energy.

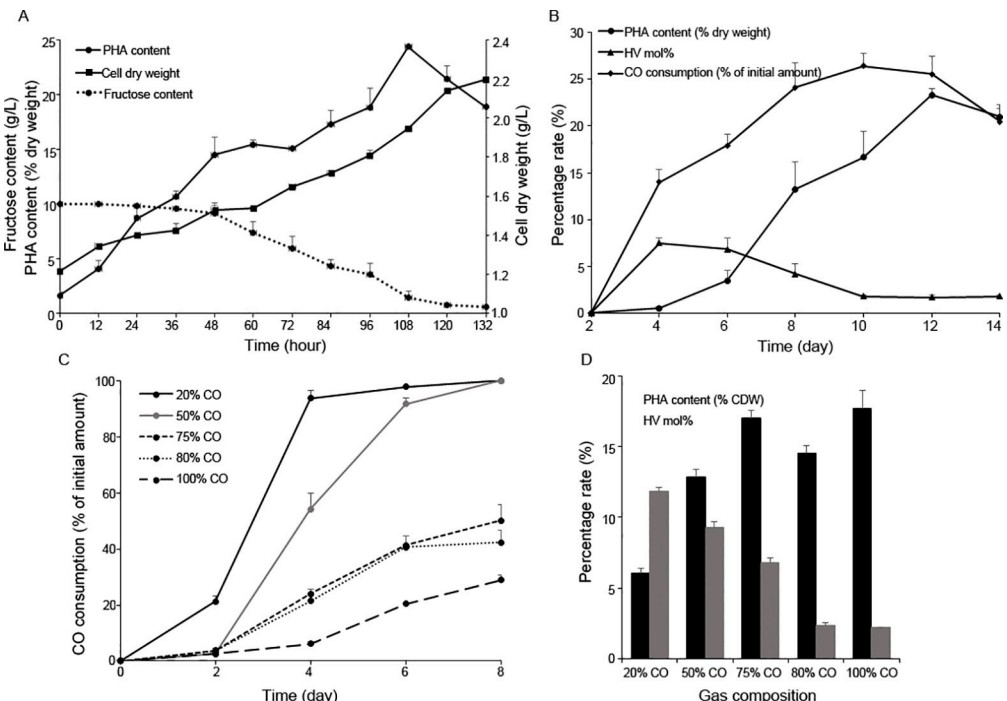

**Fig 1. PHA fermentation analysis of *R. rubrum* cultivated under different growth conditions.** (A) Growth of *R. rubrum*, PHA production and fructose uptake rate during 10g/l fructose fermentation process; (B) CO uptake rate, PHA content and 3-HV monomers proportion during pure CO fermentation process; CO uptake rate (C), PHA content and 3-HV monomers proportion (D) under different hydrogen addition ratio. Data are mean ±SD, n = 3.

After 132 h of incubation, the cultures with 10 g l$^{-1}$ fructose had grown to the highest concentration (2.2 g l$^{-1}$) (Fig 1A). And the highest PHA content of 24.4% (w/w) was detected at 108 h. Moreover, its species of PHA was PHBV which was composed of approximately 53.5% 3HB and 46.5% 3HV monomers. Interestingly, although the biomass was maintained at the initial value of 1.2 g l$^{-1}$ in the CO cultivation, a similar maximum PHBV production (23.3%) was obtained after 12 d (Fig 1B). However, the component analysis shown that the content of 3HV in this PHBV was only 2.2%, which was much lower than that on fructose medium. Furthermore, we monitored the CO consumption during gas fermentation and found the CO conversion was around 25%, which is similar to the report of Karmann et al [14].

Syngas is rich in CO and H$_2$ and its CO/H$_2$ ratio varies depending on its origin. Therefore, different syngas mixtures were applied to investigate whether variation in CO/H$_2$ ratio offers an opportunity to optimize the PHA production and CO assimilation efficiency of *R. rubrum*. We added diversity proportions of H$_2$ to achieve different proportional syngas ranging from 100% CO to 20% CO and used it as the gas phase for *R. rubrum* cultivation. Then, the CO consumption ratio was monitored. As shown in Fig 1C, the addition of H$_2$ could facilitate the uptake of CO. The CO absorption speed was largely accelerated by adding 20% hydrogen and the CO conversion was enhanced to 42%. Moreover, when the H$_2$/CO ratio was increased to 50%, CO was completely assimilated after 8 d of incubation. We speculated that the initial addition of H$_2$ was oxidized to provide energy for cells by hydrogenase, and further improved energy conversion efficiency in *R. rubrum* cells. Despite improved CO utilization, the maximum PHA concentration was gradually decreased with the increase of H$_2$ addition after 8 d of fermentation (Fig 1D). Interestingly, a significantly higher molar proportion of 3HV was observed with H$_2$ addition, which was reached 11.8% in the presence of 80% H$_2$, suggesting

that cells were growing with different metabolic strategy. Consequently, the regulation of a syngas with the desired $CO/H_2$ ratio will be a key factor to optimize PHA production in *R. rubrum*. What's more, the effects of $H_2$ also need to be in-depth assessed.

## Global changes in gene expression during fructose, CO and $CO/H_2$ fermentation processes

To get a comprehensive view of CO assimilation into biomass, we monitored gene expression changes in *R. rubrum* cultivated with an initial gas atmosphere of 100% CO or 50% CO (50% $H_2$) using RNA sequencing. 12 cell samples under three fermentation conditions (including $10g\ L^{-1}$ fructose, 100% CO, 50% CO induced fermentation and pre-culture cultivation samples, named R2YG, R2YC, R2YH and R2K respectively) were collected and subjected to Illumina high-throughput sequencer. After cleaning and filtering out low quality and ambiguous reads, 91 million clean reads containing 13.6 Gb of valid data were acquired (Table 1). The sequencing data were deposited in the National Center of Biotechnology Information (NCBI) database (accession number: PRJNA943070).

Comparisons of the expression profiles under different fermentation conditions revealed substantial differences. Compared with the cell growth stage, fructose induced medium led to up-regulation of 1384 genes, whereas 1222 genes were found to be down-regulate (Fig 2A). The highly gene modification suggest that microbial cells trigger different metabolisms to response nutrient stress. Similarly, the numbers of differential expression genes (DEGs) were found to be up to 2731and 2554 after induction culture with pure CO and 50% CO, respectively. Furthermore, pairwise comparisons were conducted among the three induction phases to identify DEGs involving in different carbon source assimilation process. Compared with 50% CO, 1744 transcripts were showed significant differential expression under 100% CO condition, of which ~50% were up regulated. By contrast, only 660 transcripts showed significant differential expression in fructose fermentation versus pure CO fermentation. This seemed to imply that the more similar metabolic pathway might be used to assimilate those two different carbon sources during PHA accumulation phase. While the presence of $H_2$ could make a different for some metabolisms of bacteria at this phase. Cluster analysis of the DEGs in a heatmap revealed that *R. rubrum* cell exhibited highly similar expression patterns under fructose and pure CO fermentation conditions (Fig 2C). However, bacterial cellular metabolism patterns under 50% CO condition were similar with control group, to a certain extent, which

**Table 1. Summary of transcription data obtained by Illumina sequencing.**

| Sample name | Raw reads | Clean reads | clean bases | Error rate (%) | Q20 (%) | Q30 (%) | GC content (%) |
|---|---|---|---|---|---|---|---|
| R2K1 | 7647650 | 7556418 | 1.1G | 0.02 | 98.06 | 94.5 | 64.58 |
| R2K2 | 7825864 | 7678970 | 1.2G | 0.03 | 97.92 | 94.2 | 64.66 |
| R2K3 | 7956798 | 7862684 | 1.2G | 0.02 | 98.04 | 94.45 | 64.73 |
| R2YC1 | 7560608 | 7443368 | 1.1G | 0.03 | 97.91 | 94.26 | 66.6 |
| R2YC2 | 7940112 | 7867056 | 1.2G | 0.03 | 96.89 | 91.48 | 65.58 |
| R2YC3 | 9028798 | 8878318 | 1.3G | 0.03 | 97.88 | 94.15 | 65.64 |
| R2YG1 | 7640584 | 7545428 | 1.1G | 0.02 | 98.03 | 94.44 | 65.31 |
| R2YG2 | 7605020 | 7518366 | 1.1G | 0.03 | 97.18 | 92.29 | 66.28 |
| R2YG3 | 8597972 | 8471042 | 1.3G | 0.03 | 97.79 | 93.91 | 64.8 |
| R2YH1 | 7717202 | 7560448 | 1.1G | 0.03 | 97.85 | 94.07 | 63.48 |
| R2YH2 | 7609948 | 7507340 | 1.1G | 0.02 | 98.04 | 94.5 | 64.44 |
| R2YH3 | 5372448 | 5219664 | 0.8G | 0.03 | 97.77 | 93.99 | 64.56 |

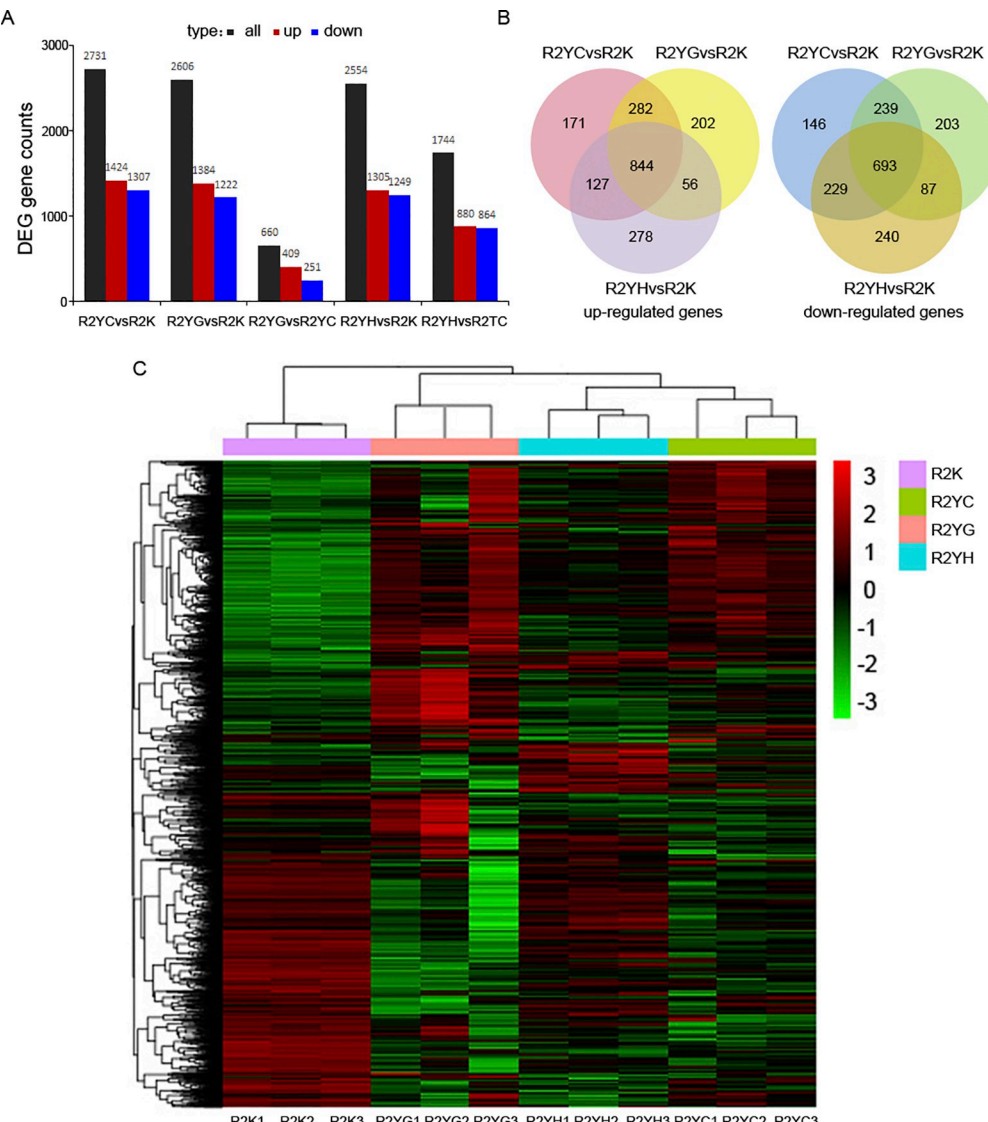

**Fig 2. Global transcriptome during fructose, CO and CO/H₂ fermentation processes.** (A) The number of differentially expressed genes in each condition; (B) The Venn diagram shows the overlap of up-regulated and down-regulated genes in various induced condition and growth stage samples. The numbers in the circles indicate the number of genes changed in each condition; (C) Dissimilar effects of fructose, pure CO and 50% CO upon global gene expression.

suggesting addition of $H_2$ might partially alleviate the stress response and restore the growth mode of *R. rubrum* cell.

## Metabolic and functional shifts in response to changing gas atmosphere

To determine the functional implication during the different fermentation stages, GO analysis was carried out on those DEGs of cell growth stage and PHA production phase. The biological functions of those DEGs were classified into three categories: molecular function, cellular component and biological process. The results showed that similar transcriptional induction under three PHA fermentation conditions were observed for the genes related to

transmembrane transport (membrane, transmembrane transport complex, plasma membrane protein complex, etc.), cellular macromolecule biosynthesis, ATP biosynthesis (ATPase activity, nucleoside-triphosphatase activity, hydrolase activity, etc.) (S1 Fig), suggesting a high degree of metabolism changes and substance exchange in response to nitrogen deficiency in the medium. Instead, a number of genes linked to peptide biosynthetic process, ribonucleoprotein complex, oxidoreductase activity, NADH dehydrogenase subunits were well expressed in the growth phase, which are consistent with their importance of cell assembly and energy conservation for cell growth. In addition, several flagellum-dependent cell motility related genes were also strongly induced, that corresponds to Raberg et al [25]. Reported that flagellation was strongly occurred during growth and stagnated during PHA synthesis.

To gain insight into the specific mechanism of *R. rubrum* under different gas atmosphere, we comparatively analyzed the cellular metabolic adaptations to pure CO and 50% CO fermentation (Fig 3A). At a first glance, it seemed to imply that the pure CO atmosphere triggered intense cellular stress response. Large amounts of DEPs annotated into cellular response to stimulus, signal transduction, ATPase activity was significantly expressed, as well as several transcription factors and regulators. Such as the genes coding for aldehyde dehydrogenase (*Rru_A0656*, *Rru_A1542*, *Rru_A0914*), iron-containing alcohol dehydrogenases (*Rru_A0904*) and several NADH dehydrogenases (e.g., *Rru_A0320*, *Rru_A0314*, *Rru_A0321*, etc.), which are belonging to the reactive oxygen species (ROS) detox system, were significantly induced (S1 Appendix). Aldehyde dehydrogenases are known as a critical component in response to various environmental stresses, particularly the ROS stress [26]. Taken together, these stress related genes reflected an intracellular oxidative stress reaction induced in pure CO atmosphere, which was consistent with the high PHA yield under CO fermentation. There is an agreement that the responses of *R. rubrum* to stress pressure are firstly triggered by the

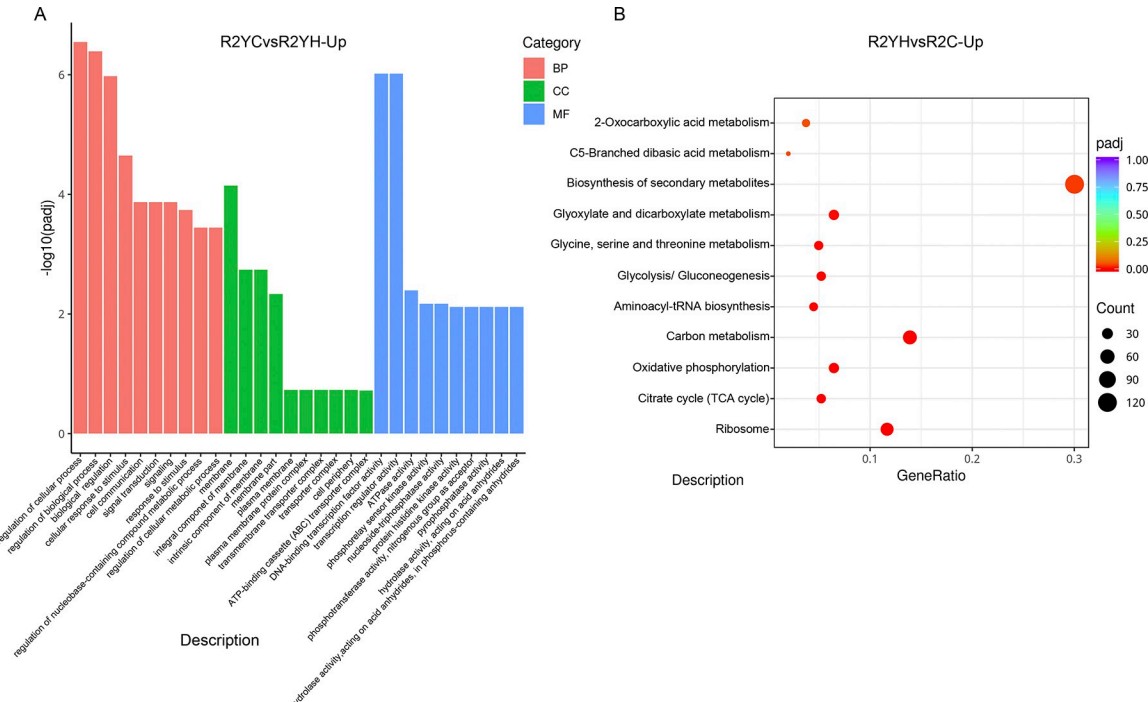

**Fig 3. Categories analysis of DEG in pure CO and 50% CO fermentation.** (A) GO biological processes in genes upregulated under pure CO fermentation; (B) Top 11 pathways enriched by KEGG analysis for upregulated genes in 50% CO fermentation.

activation of ROS signal transduction pathways and PHA biosynthesis plays a key role in balancing the redox state. In addition, comparing with cell growth phase, genes enriched in glycolysis/gluconeogenesis, citrate cycle (TCA cycle), carbon metabolism pathways were significantly downregulated in pure CO fermentation stage, which was corresponds with the stunted bacterial growth (S1 Appendix). Interestingly, hydrogen addition significantly induced protein biosynthesis process and the expression of genes linked to translation, peptide biosynthetic process, tRNA aminoacylation for protein translation and ribonucleoprotein complex. All these changes suggested a strong induction in protein synthesis might lead to a gradual recovery of the growth restarted. Combined with KEGG pathway enrichment analysis, we picked 11 pathways enriched by DEPs (padj<0.05) (Fig 3B). Most of the $H_2$ up-regulated genes were enriched in the pathways of ribosome, carbon metabolism and biosynthesis of secondary metabolism (59%, total 224 genes) (S2 Appendix). Apart from these three pathways, the remaining pathways (e.g., citrate cycle (TCA cycle), glycolysis/gluconeogenesis, glyoxylate and dicarboxylate metabolism, C5-Branched dibasic acid metabolism, etc.) were almost related to carbon metabolism and carbon storage. These results imply that carbon metabolism can act as the central response to the change of fermentation gas atmosphere.

## Metabolic pathways of CO assimilation and PHA production in pure CO and 50% CO fermentation

**(i) CO assimilation and $H_2$ production as a first step of the *R. rubrum* gas fermentation.** During gas fermentation, $H_2$ addition accelerated the CO absorption and elevated molar proportion of 3HV in PHAs, although PHA concentration was decreased. To gain insight into the specific mechanism of CO assimilation into biomass under different gas atmosphere, an expression analysis of genes potentially involved in the assimilation/fixation of CO was carried out. *R. rubrum* can use CO under anaerobic conditions as carbon and energy source. When exposed to CO, CODH is induced to oxidate CO into $CO_2$. Evaluation of expression profiles of the genes coding for CODH (*Rru_A1427*) and its key regulator CooA (CO-sensing transcription activator, *Rru_A1431*) revealed that transcripts of *CODH* and *CooA* were more abundant at gas fermentation phase compared to cellular growth phase (Table 2). During the CO oxidation to $CO_2$, two protons are reduced to $H_2$ catalyzed by a CO-tolerant membranous energy-conserving [NiFe]-hydrogenase (ECH). Then, $H_2$ is released as a co-production of the water gas-shift reaction. Consistently, the genes coding for CO-linked ECH (*Rru_A1425*), a formate-linked hydrogenase (*Rru_0326*) and a $H_2$ uptake hydrogenase (*Rru_A1161*) were found up-regulated in different degree upon CO exposure. To get a comprehensive view of CO assimilation, the expression levels of *CODH* and *CooA* were monitored by RT-PCR at different time points under pure CO and 50% CO conditions. As shown in Fig 4, the expression levels of *CODH* and *CooA* were upregulated in gas fermentation process about 100-fold and sevenfold respectively, indicating a strong upregulation of *CODH* and *CooA* by CO. In agreement with the phenotypic observations, the expression of *CODH* and *CooA* was significantly higher in the presence of 50% $H_2$ than that in pure CO at the initial stage of fermentation. Moreover, the expression of *CODH* and *CooA* was exhibited a gradual decrease with the fast CO consumption after 8 days of fermentation. This observation suggests that 50% CO fermentation may have more advantages than pure CO fermentation, at least from the perspective of gas utilization efficiency.

**(ii) Identification of potential routes involved in $CO_2$ fixation into organic compounds during different gas fermentation.** Subsequently, those $CO_2$ generated by oxidation is fixated into organic compounds via various carboxylases. Our study indicated that *R. rubrum* adopted different carboxylation strategies under different gas fermentation conditions, thus

**Table 2. Genes involved in CO assimilation, CO₂ fixation and redox homeostasis highlighted by transcriptome analysis.**

| Gene name | Description Role in *R. rubrum* | Pure CO induction vs. growth phase | | 50% CO induction vs. growth phase | |
|---|---|---|---|---|---|
| | | p-value | log2 Fold change | p-value | log2 Fold change |
| **CO assimilation** | | | | | |
| *Rru_A1427* | Carbon-monoxide dehydrogenase | 5.91E-86 | 6.089 | 1.39E-83 | 4.606 |
| *Rru_A1431* | Crp/Fnr family transcriptional regulator | 6.66E-21 | 2.681 | 6.35E-07 | 1.324 |
| *Rru_A1425* | ECH hydrogenase subunit E | 1.76E-93 | 5.544 | 4.74E-87 | 4.161 |
| *Rru_A0326* | transcriptional regulator | 1.64E-09 | 1.091 | ND[a] | ND |
| *Rru_A1161* | Ni-Fe hydrogenase | 1.53E-25 | 3.968 | 1.11E-11 | 2.518 |
| **Calvin-Benson-Bassham cycle** | | | | | |
| *Rru_A2400* | Ribulose bisphosphate carboxylase | 0.015 | 1.137 | ND | ND |
| **methylmalonyl-CoA (MMC) pathway** | | | | | |
| *Rru_A0052* | Biotin carboxylase | ND | ND | 1.18E-90 | 2.350 |
| *Rru_A0053* | Carboxyl transferase | ND | ND | 2.41E-83 | 2.305 |
| *Rru_A2318* | 2-methylcitrate dehydratase | 1.72E-49 | 5.708 | 4.62E-28 | 3.536 |
| *Rru_A2320* | Phosphoenolpyruvate phosphomutase | 6.41E-28 | 4.305 | 6.84E-11 | 1.857 |
| *Rru_A2319* | 2-methylcitrate synthase/Citrate synthase | 4.49E-15 | 5.863 | 8.30E-11 | 2.855 |
| **ethylmalony-CoA (EMC) pathway** | | | | | |
| *Rru_A1201* | Mesaconyl-CoA hydratase | ND | ND | 0.00013 | 0.8 |
| *Rru_A3062* | Methylmalonyl-CoA mutase | 1.63E-08 | 1.338 | 1.21E-16 | 1.947 |
| *Rru_A3063* | Crotonyl-CoA carboxylase/reductase | 2.38E-08 | 1.800 | 3.65E-54 | 3.723 |
| *Rru_A3064* | Isovaleryl-CoA dehydrogenase: Acyl-CoA dehydrogenase | ND | ND | 4.74E-11 | 1.429 |
| **pyruvate ferredoxin oxidoreductase (PFOR)** | | | | | |
| *Rru_A2398* | Pyruvate- flavodoxin oxidoreductase | ND | ND | 0.002392 | 0.561 |
| **Threonine dependent branched-chain amino acid (BCAA) biosynthesis pathways** | | | | | |
| *Rru_A1135* | Aminotransferase | ND | ND | 0.003836 | 0.325 |
| *Rru_A0743* | Aspartate kinase | ND | ND | 0.013116 | 0.284 |
| *Rru_A1196* | Aspartate semialdehyde dehydrogenase | ND | ND | 2.20E-34 | 2.005 |
| *Rru_A3053* | Homoserine kinase | 1.21E-06 | 0.938103 | ND | ND |
| *Rru_A0467* | Acetolactate synthase large subunit | ND | ND | 5.35E-20 | 1.689 |
| *Rru_A0468* | Acetolactate synthase small subunit | ND | ND | 0.005039 | 0.561 |
| **branched-chain amino acid (BCAA) degradation pathways** | | | | | |
| *Rru_A1977* | Pyruvate ferredoxin/flavodoxin oxidoreductase | 1.95E-24 | 4.686015 | 6.32E-17 | 3.731 |
| *Rru_A1978* | Indolepyruvate ferredoxin oxidoreductase | 4.37E-56 | 3.829587 | 1.13E-45 | 3.322 |
| **TCA** | | | | | |
| *Rru_A2721* | 2-oxoglutarate synthase | ND | ND | 0.011457 | 0.342 |
| *Rru_A2129* | fumarase | ND | ND | 3.09E-13 | 1.052 |
| **PHA** | | | | | |
| *Rru_A0278* | Transcriptional regulator | ND | ND | 7.46E-11 | 1.442 |
| *Rru_A1816* | Poly(R)-hydroxyalkanoic acid synthase (PhaC3) | 1.55E-07 | 1.052 | ND | ND |
| *Rru_A2111* | Hypothetical phasin protein | ND | ND | 1.32E-05 | 0.746 |
| *Rru_A2817* | Phasin protein | 9.57E-12 | 1.795 | 1.66E-18 | 1.322 |
| *Rru_A1585* | Polyhydroxyalkanoate depolymerase PhaZ1 | 2.62E-07 | 0.921 | 3.14E-17 | 1.861 |
| *Rru_A3356* | Polyhydroxyalkanoate depolymerase | 2.40E-06 | 1.590 | 5.57E-06 | 1.267 |

[a] Not detected

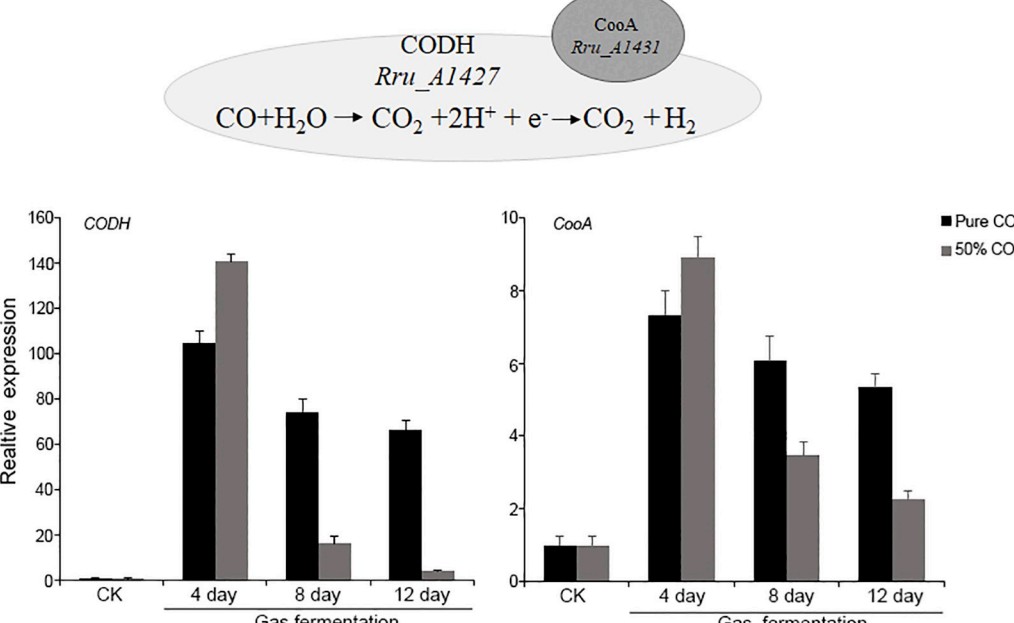

**Fig 4. Changes of *CODH* and *CooA* expression during different gas fermentation process.**

affecting the distribution of intracellular components. Related carboxylases were compiled in Table 2. The ribulose bisphosphate carboxylase (RuBisCO), the main enzyme of the Calvin-Benson-Bassham (CBB) cycle, showed a significantly higher abundance in pure CO induced fermentation (*Rru_A2400*, p = 0.015, fold change = 1.137), which implicated the necessity of CBB cycle for $CO_2$ incorporation to central metabolites upon CO exposure. Moreover, the CBB pathway was also used as a redox balancing pathway to dissipate the reduced equivalents generated during 100% CO induced fermentation [27, 28]. Besides Rubisco other carboxylases belonging to enthylmalony-CoA (EMC) pathway (*Rru_A3062*, p = 1.63E-08, fold change = 1.338; *Rru_A3063*, p = 2.38E-08, fold change = 1.800) also participates in the assimilation of catalytic $CO_2$. Those are consistent with the previous reports that the EMC pathway acts as a $CO_2$ assimilation and redox balancing route for acetate growing cells [10, 21].

However, in contrast to what was observed in the pure CO fermentation, the addition of $H_2$ make a great change to the carbon assimilation and metabolism pathway. The transcriptomic data and RT-PCR expression assay both revealed that the expression level of *RuBisCO* showed no variation between growth phase and 50% CO fermentation phase, which implied the CBB pathway was not essential for $CO_2$ fixation under 50% CO gas atmosphere (Fig 5). Significantly, a higher abundance of genes belonging to branched-chain amino acid (BCAA; Ile, Leu, Val) biosynthesis and degradation pathways were found to by transcriptome analysis, which represent another alternative route to managing the redox balance (*Rru_A0467*, p = 5.35E-20, fold change = 1.689; *Rru_A0468*, p = 0.005, fold change = 0.561; *Rru_A1977*, p = 6.32E-17, fold change = 3.731; *Rru_A1978*, p = 1.13E-45, fold change = 3.322) (Fig 6). These results implied the potential involvement of ILV biosynthesis in the metabolism of syngas (CO/$H_2$) in *R. rubrum*. According to this clue, we discovered that the reversal tricarboxylic acid (TCA) cycle was operated, specifically, α-ketoglutarate (αKG) synthase (*Rru_A2712*, p = 0.011, fold change = 0.342) which oxidizes succinyl-CoA to form αKG and meanwhile fixes one molecule of $CO_2$. This pathway and the subsequent synthesis of αKG-derived amino acids were

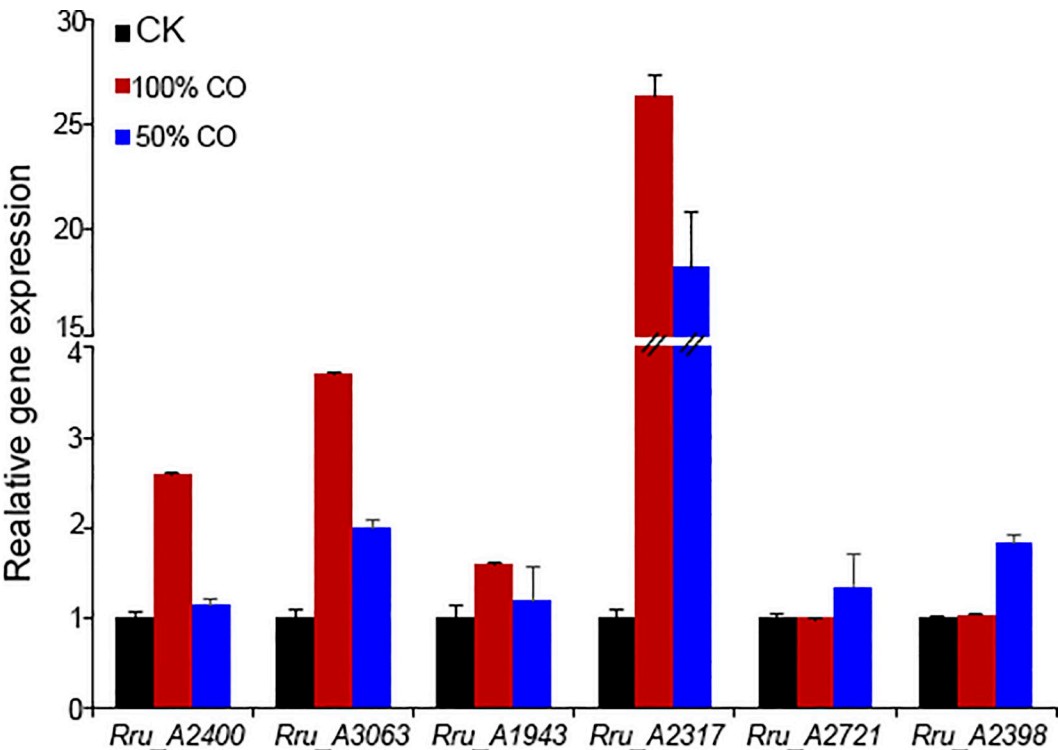

**Fig 5. Expression analysis of CO$_2$ fixation related genes under different gas growth condition by RT-PCR.**

previously proposed as an important mechanism for electron balance in a *R. rubrum* Calvin-cycle mutant [29, 30]. Moreover, the fumarase which plays a key role in the TCA cycle to catalyze the reversible dehydration of fumaric acid to malate also showed a massively higher relative abundance in syngas fermentation condition (*Rru_A2129*, p = 3.09E-13, fold change = 1.052). Its enzymatic products fumarate/malate are important precursors for the threonine-dependent isoleucine synthetized [27, 30]. In addition, our RT-PCR analysis revealed that two other carboxylases related to TCA cycle were also highly upregulated in gas fermentation stage. Propionyl-CoA carboxylase (Rru_A1943) catalyzed propionyl-CoA yielding methylmalonyl-CoA linking to succinyl-CoA synthesis and pyruvate carboxylase (Rru_A2317) catalyzed pyruvate yielding oxaloacetate (Fig 5).

Currently, isoleucine pathway has always been considered as a reduced equivalent consuming pathway. Very interestingly, our data also revealed that a series of enzymes in reductive threonine-dependent pathway were found to be upregulated in the syngas condition (*Rru_A1135*, p = 0.003, fold change = 0.325; *Rru_A0743*, p = 0.013, fold change = 0.284; *Rru_A1196*, p = 2.20E-34, fold change = 2.005). One oxaloacetate and one pyruvate were used to initiate the threonine-dependent pathway and finally product threonine which was subsequently catalyzed by deaminase to form oxobutanoate. Then the acetolactate synthase (Rru_A0467; Rru_A0468) are responsible for the catalysis of oxobutanoate to start the BCAA biosynthesis pathway. Those results highlighted the cellular preferred the reductive threonine-dependent pathway for isoleucine synthesis instead of the citramalate-dependent pathway. Additionally, pyruvate is another key metabolite of the BCAAs biosynthesis [31]. The pyruvate ferredoxin oxidoreductase (PFOR) catalyzes a reversible reaction and can combine CO$_2$ with acetyl-CoA to form pyruvate. Coherently, PFOR coded by the *Rru_A2398*, were observed to be

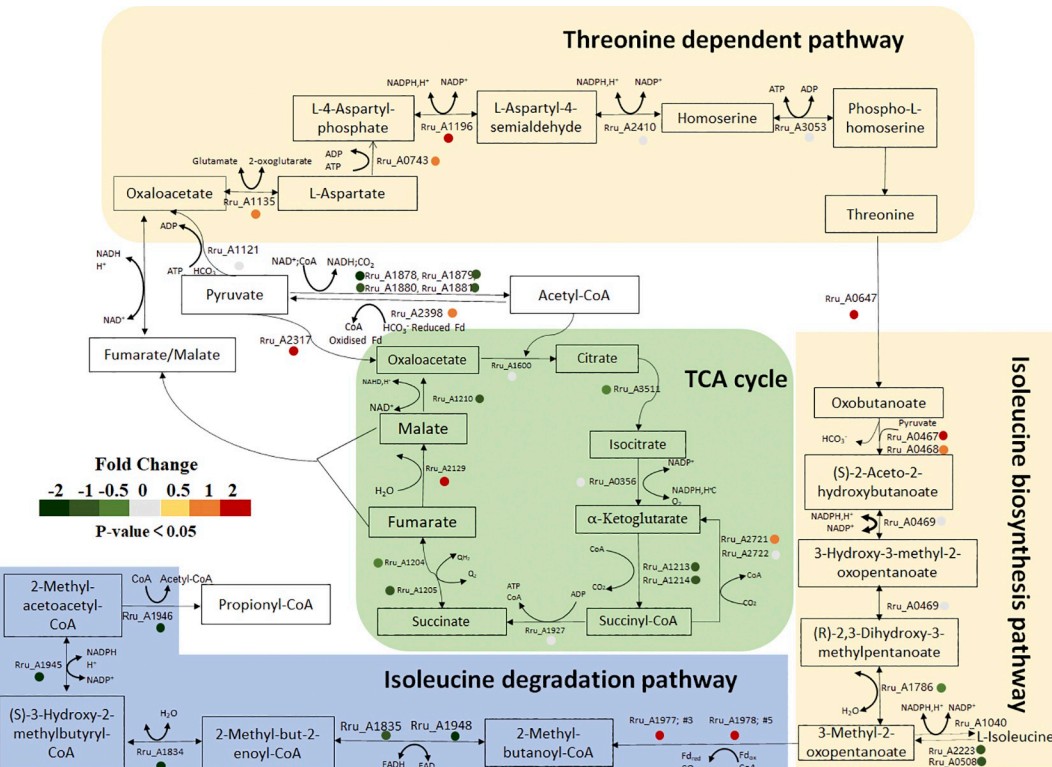

**Fig 6. Schematic representation of the central carbon metabolism and threonine-dependent isoleucine biosynthesis pathways in 50% CO fermentation condition, as highlighted by transcriptome data.** The colored circles indicate the fold changes between 50% CO fermentation and growth condition, ranging from red (genes are more abundant in the 50% condition) to green (genes are less abundant in the 50% condition). Nonsignificant genes are represented by grey circle.

more abundant in 50% CO fermentation condition, while no differences were found under pure CO atmosphere. And as in previous study, starting from acetyl-CoA, *de novo* synthesis of BCAA consumes reduced ferredoxin and NADPH contributing to cellular electron balance [30, 31]. These all reflected that the BCAA biosynthesis pathway and its related metabolic processes TCA cycle represent a key way for $CO_2$ fixation and redox potential maintenance in *R. rubrum* during syngas fermentation process. To further validate this assertion, we analyzed the accumulation of intracellular amino acids in *R. rubrum* growing in pure CO or syngas conditions. For the concentrations of BCAAs analysis, results showed that only leucine was detected and significantly different in these two fermentation conditions (S3 Appendix). As previous study showed that the synthesis of BCCAs was impacted by various environmental changes and the accumulation of isoleucine-leucine-valine (ILV) was transient [32]. Further detailed experimentation will be required to reveal the accumulation regular of BCAAs and in-depth understand their role in redox state balance. Besides, other amino acids also exhibited obvious changes in concentration. In accordance with previous studies, the synthesis of these amino acids via specific carboxylation reactions (e.g., Glu derives from αKG, Ser from 3-phospho-glycerate, Ala from pyruvate, etc.) could also contributed to electron balance [30].

**(iii) The production of the copolymer poly(3-hydroxybutyrate-co-3-hydroxyvalerate).** Short-chain-length (SCL) PHAs (monomer units of C3 to C5) are synthesized from acetyl-CoA via three enzymatic reactions by three different enzymes. The first reaction is the condensation of two acetyl-CoA molecules into acetoacetyl-CoA by β ketothiolase (PhaA). In the

second reaction, acetoacetyl-CoA reductase (PhaB) reduces the acetoacetyl-CoA to form 3-hydroxyalkanoate-CoA. Finally, PHA synthase (phaC and phaE) catalyzes the polymerization of 3-hydroxyalkanoate-CoA monomers to form a polyhydroxyalkanoate. The gas substrate used to fermentation and its assimilation pathway directly impact the production and composition of the PHA polymers. Our PHA quantitation analysis revealed a higher production of PHAs in the pure CO fermentation than that of syngas. However, transcriptomic data revealed that there were no significant differences in the expression of the three key enzymes of PHA synthesis pathway (phaA, phaB and phaC; separately coded by *Rru_A0274*, *Rru_A0273*, and *Rru_A0275*) under both fermentation conditions. Interestingly, one of the homologous PHA polymerase gene *phaC3* (*Rru_A1816*) and a phasin (*Rru_2817*) associated with PHA granules were upregulated in pure CO fermentation. In case of syngas cultivation, the regulator of the phaCAB cluster (*Rru_A0278*) and two phasins (*Rru_2111*, *Rru_2817*) were shown more abundant. Meanwhile, two depolymerases (*Rru_A1585* and *Rru_A3356*) were highly upregulated under both conditions. The enzyme of polymerization and depolymerization of PHA work together to determine the final production of PHA. Those above gene expression changes were influenced by the specific time points of sampling. Samples of 8th day gas fermentation were selected for RNA sequencing. And at this time point the cellular PHA formation might reach to platform stage, especially in syngas fermentation. Moreover, in the late fermentation, PHA would undergo a rapid mobilization with the shortage of carbon sources. Since transcriptome assay only reflected the expression of PHA synthesis-related gene at a specific time, we further analyzed the expression of the serval key enzymes of PHA synthesis pathway at different fermentation times. RT-PCR analysis showed that the *Rru_A0275*, *Rru_A1816* and *Rru_A0274* transcript levels were elevated after 4 days in pure CO culture, whereas *Rru_A0273* gene expression was slightly decreased (Fig 7). Moreover, the expression of *Rru_A1816* and *Rru_A0274* were evident at least 8 days, which is consistent with the gradually increased intracellular PHA accumulation during pure CO fermentation. Unlike 100% CO fermentation, only the PHA polymerase coding gene *Rru_A0275* and *Rru_A1816* were induced and reached the highest spot on the fourth day after transferred into syngas medium and then shown a gradually decreasing trend, which might finally lead to a decrease in PHA production. We speculated that this might be due to the rapid consumption of CO during 50% CO fermentation process. With the shortage of carbon sources PHA would undergo a rapid mobilization.

PHA production has been considered as a mechanism to maintain the redox homeostasis of the cell and its biosynthesis is induced by the redox stress [27, 33]. Indeed, there are other alternative pathways in *R. rubrum* competing with PHA synthesis to help balance the redox pool state. Moreover, studies show that PHA production may be blocked while another

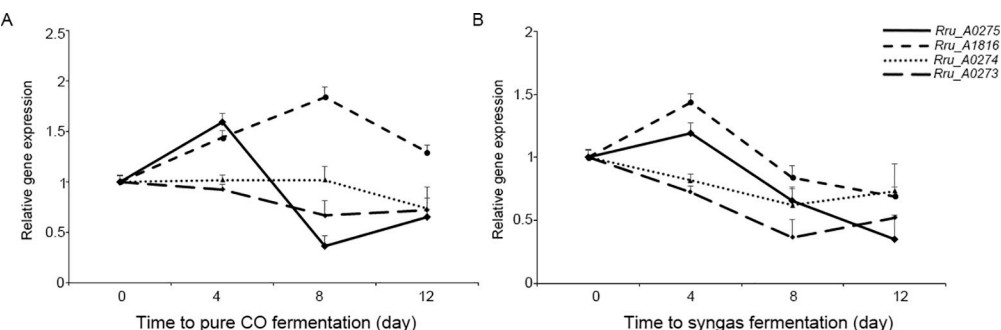

**Fig 7. Expression analysis of PHA biosynthesis related genes during 100% CO and 50% fermentation process by RT-PCR.**

electron sink metabolism is available. We proposed that under syngas fermentation condition, the BCAA biosynthesis pathway might act as a key electron-sinking pathway to bypass PHA production, and then eventually lead to the decrease of intracellular PHA accumulation. Even though, besides the PHA polymer yield improvement, the physicochemical properties and biodegradability optimizing, which are dependent on their monomeric constituents and relative order in the polymer, are other key points for the PHA research [34, 35]. As our phenotypic characteristics result shown, $H_2$ addition induced a significantly higher molar proportion of 3-HV, which triggers our more consideration about gas fermentation for PHA production. Previous research showed that the 3-HV monomers of the copolymer (3-hydroxybutyrate-co-3-hydroxyvalerate) might originate from the condensation of acetyl-CoA and propionyl-CoA into 3-ketovaleryl-CoA and the odd number of carbon sources, such as propionate, could induce the production of the 3-HV content [36, 37]. Moreover, the pyruvate synthesized by PFOR could also be converted to 3-HV precursor molecule propionyl-Co-A. Meanwhile the biosynthesis and degradation pathways of L-isoleucine (ILV) could convert acetyl-CoA into propionyl-CoA, which constituted a new photoheterotrophic carbon assimilation pathway. These results suggested that the production of PHA might be modified by the suitable addition of $H_2$ for the desired copolymer synthesis.

## Fed-batch cultivation with *R. rubrum* on syngas

The results above revealed that $H_2$ addition could accelerate the CO absorption and improve gas utilization efficiency in *R. rubrum*. However, due to the activation of the ILV pathway which partly replaced PHA biosynthesis, the final production of PHA was decreased. In the actual fermentation process, we should comprehensively consider fermentation rate and PHA yield. With the above factors, we regulated the CO:$H_2$ ratio at 3:1 (simulated syngas) as initial gas feed and compared it with the pure CO fermentation. A fed-batch bioprocess in a 3-l bioreactor was performed and the final broth composition, regarding biomass production, PHA content and acetate concentration were monitored for each culture (Fig 8). Similarly,

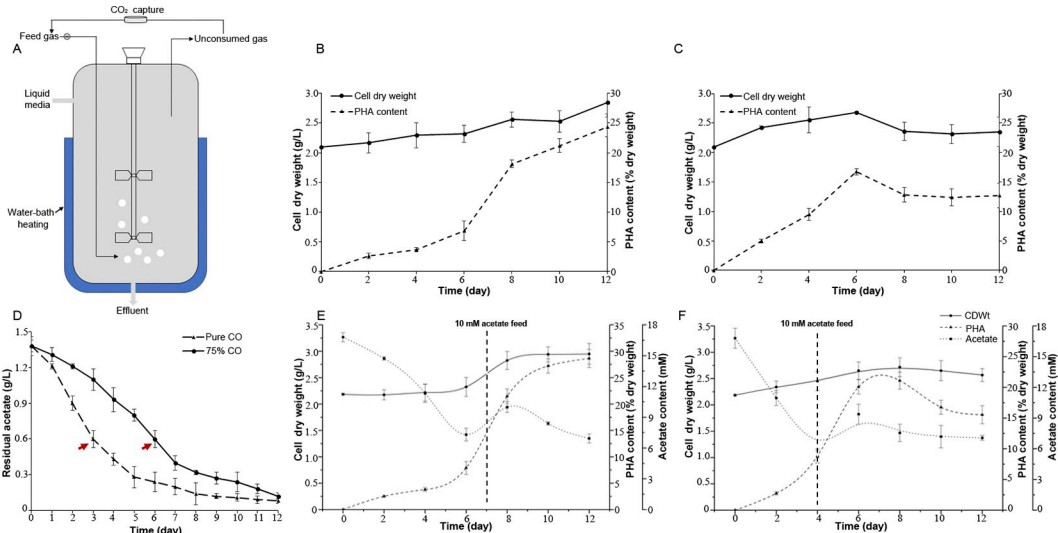

**Fig 8. Fed-batch fermentation of *R. rubrum* in a 3-l bioreactor.** (A) Schematic layout of the bioreactor for gas fermentation; Biomass concentrations and PHA yields under different gas conditions: (B) pure CO fermentation, (C) 75% CO fermentation; (D) Acetate concentrations during different fermentation conditions; After10mM acetate added in the mid-stage, biomass concentrations, PHA yields and acetate content in pure CO condition (E), or 75% CO condition (F). Data are mean ±SD, n = 3. acetate-feeding strategies.

*R. rubrum* was cultivated by the two-step method as described in Methods. To achieve rapid growth, the cells were first cultured in broth medium to the stationary phase, and then transferred to the 3-l bioreactor with RRNCO mineral medium (containing 1.38 g l$^{-1}$ sodium acetate as co-substrate) for gas fermentation experiments.

The most remarkable change occurred with 25% H$_2$ added was a significant increase in PHA accumulation rate, yielding the maximum PHA content (16.7% ww$^{-1}$) after 6 days of fermentation (Fig 8C). During pure CO growth, although its final PHA yield up to 24.3% ww$^{-1}$, its fermentation time was extended to 12 days (Fig 8B). In addition to the gas substrate, acetate in liquid culture also has a positive effect on PHA accumulation. Revelles et al. reported acetate providing the carbon skeleton for PHAs and promoting PHA accumulation, and therefore, the effect of acetate during gas fermentation was assessed [10]. As shown in Fig 8D, the initial acetate was gradually consumed along with the extension of fermentation time. Differently, the maximum values of acetate consumption rates (0.3 g l$^{-1}$d$^{-1}$) were observed 3 days after inoculation in CO/H$_2$ atmosphere, and acetate content was decreased to 31% on the 4$^{th}$ day of cultivation (Table 3). At this point, only 24% of CO was consumed from the initial syngas which implied that the accumulation of PHA was tightly related to acetate concentration. While in pure CO condition, acetate content was decreased to 43% on the 6$^{th}$ day of cultivation and 20% of CO was conversed at this moment. In this sense, the initial H$_2$ addition indeed excited carbon sources utilization (including CO assimilation and acetate conversion) and in part accelerated PHA accumulation. Although the cellular PHA was gradually accumulated during gas fermentation process, the biomass yield had not significant increase. This might be due to the period of bacterial cell growth. The stationary phase cells were used for gas fermentation in our experiment. In this condition, as Godoy et al. reported polymer accumulation prevailed over residual biomass formation [38]. Therefore, accompanied by PHA accumulation biomass formation had a slight increase at earlier part of fermentation, and at the end of the fermentation with the mobilization of PHA biomass yields shown a downward trend.

Considering the high consumption rate of acetate in *R. rubrum* gas fermentation process, acetate-feeding strategies were established to optimize the PHA production process. And 10 mM acetate was respectively added on the 4$^{th}$ day of CO/H$_2$ cultivation when the initial residual acetate amount was only 31% or the 7$^{th}$ day of pure CO condition with 29% residual acetate (Fig 8E and 8F). With a controlled acetate feeding, cells re-started PHA formation at a faster rate, eventually reaching platform stage. The most remarkable results were the highly efficient accumulation of PHA in case of CO/H$_2$ cultivation, yielding the maximum PHA production (PHBV around 20% ww$^{-1}$, with 6%mol 3HV) in only six days (Table 3). Compared with pure CO condition, its fermentation period was decreased by half, although its PHA production was not as high as that of the pure CO cultivation (PHBV around 28% ww$^{-1}$, with 2% mol

**Table 3. General profiling of *R. rubrum* grown in a 3-l bioreactor under different gas compositions.**

| Fed-batch | Cultivation days | CDW (g l$^{-1}$) | PHBV (% ww$^{-1}$) | HV monomer composition (%$^a$) | CO consumption (%$^b$) | Acetate consumption (%$^c$) |
|---|---|---|---|---|---|---|
| **100% CO** | 6 | 2.32±0.18 | 7.90±1.24 | 2.2±0.29 | 20.38±0.03 | 56.52±1.40 |
| | 12 | 2.95±0.19 | 28.64±1.74 | 2.4±0.17 | 62.86±1.62 | |
| **75% CO** | 4 | 2.45±0.18 | 8.8±1.17 | 6.3±0.23 | 24.06±1.57 | 68.84±0.62 |
| | 6 | 2.65±0.17 | 20.09±1.17 | 6.2±0.27 | 50.20±5.57 | |

Values represent the mean ± standard deviation of three independent biological replicates. The cell dry weight (CDW), PHA (% cell dry weight), CO and acetate consumptions during fermentation process (100% CO and 75% CO gas atmosphere) were determined.

$^a$ Percentage of HV in polymer (mol%)

$^b$ Percentage of gas conversion (%) from the initial concentration on syngas.

$^c$ Percentage of acetate absorption (%) from the initial concentration on RRNCO media.

**Table 4. Comparative PHA production in *Rhodospirillum rubrum* under different fermentation conditions.**

| Growth conditions | Cultivation modes | Carbon sources | PHA yields (% ww$^{-1}$) and detection time-point | References |
|---|---|---|---|---|
| 550 R8AH, light, anaerobic, N limitation | Photo-heterotrophic | Yeast extract (1g l$^{-1}$) + β-Hydroxybutyrate (30 mM) | PHB: 46.8<br>10 days | [39] |
| | | Yeast extract (1g l$^{-1}$) + Acetate (30 mM) | PHB: 19.2<br>10 days | |
| | | Yeast extract (1g l$^{-1}$) + Propionate (30 mM) | PHBV: 2.2<br>(HV: 58.5%) [a]<br>10 days | |
| RRNCO (with 10 g l$^{-1}$ fructose), dark, not strictly anaerobic, N limitation | Chemo-heterotrophic | Fructose (10 g l$^{-1}$) | PHBV: NM[b]<br>(HV: 46.5%)<br>NM[b] | [40] |
| RRNCO (with fructose, 1 g l$^{-1}$ yeast extract), dark, aerobic-anaerobic transition, N limitation | Chemo-heterotrophic | Yeast extract (1 g l$^{-1}$) +<br>Aerobic condition: fructose (13.3 mM)<br>Anaerobic condition: fructose (40 mM) | PHBV: 29<br>(HV: 75%)<br>4 days | [38] |
| | | Yeast extract (1 g l$^{-1}$) +<br>Aerobic condition: fructose (13.3 mM)<br>Anaerobic condition: fructose (40 mM)<br>+ bicarbonate (12 mM) | PHBV: 81<br>(HV: 86%)<br>12 days | |
| RRNCO (with yeast extract), light, anaerobic, N limitation | Photo-heterotrophic | Yeast extract +<br>Artificial gas (56.0% N$_2$, 17.2% CO, 16.3% CO$_2$, 8.8% H$_2$) | PHA: 34.8<br>6 days | [42] |
| RRNCO, dark, anaerobic, N limitation | Chemo-heterotrophic | Syngas (50% CO and 50% N$_2$) + Acetate (15 mM) | PHB:26<br>9 days | [23] |
| RRNCO, light, anaerobic, N limitation | Photo-heterotrophic | Syngas (40% CO, 40% H$_2$, 10% CO$_2$ and 10% N$_2$) + Acetate (10 mM) | PHB: 28<br>NM[b] | [10] |
| RRNCO (with 15mM fructose), light, anaerobic, P limitation | Photo-heterotrophic | syngas (25% CO, 25% H$_2$, 5% CO$_2$, and 45% N$_2$) + Acetate (10 mM) + fructose (15 mM) | PHB: 30<br>NM[b] | [43] |
| RRNCO, light, anaerobic, N limitation | Photo-heterotrophic | CO (initial ppCO 0.6 bar) + Acetate (10 mM) | PHB: 8.3±3.4<br>12 days | [24] |
| RRNCO, light, anaerobic, N limitation | Photo-heterotrophic | CO + Acetate (16.8 mM) | PHBV: 28.6 (HV:2.4)<br>12 days | This work |
| | | syngas (75% CO and 25% H$_2$) + Acetate (16.8 mM) | PHBV:20 (HV:6.2)<br>6 days | |

[a] Monomers found in polymer (mol%)

[b] Not mentioned

3HV). These results suggested that H$_2$ plays an important role during the start-up stage of gas fermentation, and it may serve as an energy supply to activate cellular carbon assimilation and accelerate PHA synthesis. Table 4 shows that there is a large variation in the PHA content of *R. rubrum*, ranging from 8% to 81%, mainly depending on the type of carbon sources and growth conditions. Using aerobic-anaerobic two-phase growth with fructose (C6) as carbon sources, co-polymer PHBV production finally reached 81% CDW (with 86% mol 3HV) in 12-days fermentation [38]. This was the highest PHBV yield and proportion of HV monomer by utilizing C6 substrate. However, the cost of the raw materials (especially carbon source) required in the mass PHA production is a main limiting factor for industrial applications [41]. Syngas fermentation offered an attractive economic prospect for PHA production. The cost of producing the PHA with syngas (C1 substrates) is not only less expensive than that by sugar but also environmentally friendly. According to Table 4, our procedures indeed increased PHA production when compared to previous works operating in the same culture medium. Our pure CO fermentation results were very close to those achieved with *R. rubrum* on syngas in continuous light or dark cultures [10, 23]. Furthermore, our PHBV productivity in syngas atmosphere is

better than others achieved with syngas or CO fermentation and achieve its maximum PHBV yield in 6 days. Therefore, it is interesting for further developing new strategies for initial syngas selection and gas composition regulation to improve PHA production of *R. rubrum*.

## Conclusions

Syngas fermentation presents an economical and effective mode for PHA production in *R. rubrum*. Deeply exploring the metabolic mechanisms of carbon assimilation under different gas environments offers an opportunity to improve growth and PHA production of *R. rubrum*. Our data support previous results indicating that *R. rubrum* could use the BCAAs biosynthesis pathway to help maintain redox homeostasis during photoheterotrophic metabolism [27, 33]. In addition to some well-described electron sinks, such as calvin cycle, $H_2$ production and PHA biosynthesis, BCAA biosynthesis has been recently highlighted as one of alternatives to those pathways. We have demonstrated here that the presence of $H_2$ is profit for the photoheterotrophic assimilation of CO and make a significant change for the cellular carbon assimilation pathway. Moreover, accompanied by the occurrence of water-gas reaction, the accumulation of co-producing $H_2$ was also optimized and opens a possibility of exploring cogeneration of bio-$H_2$ during syngas fermentation. Based on transcriptome comparative analysis, we revealed that the addition of $H_2$ triggered a series of BCAA related metabolism instead of CBB cycle to regulate the redox balance. To date, PHA industrial production from syngas has been limited by the variability of syngas and the insufficient productivities of PHA. This finding has important implication for the biotechnological use of *R. Rubrum* for PHA production on syngas. $H_2$/CO ratio in syngas might act as a key adjuster knob to balance the cellular various metabolisms to regulate the PHA productivity. In the actual application process, with only a few adaptations ($H_2$/CO ratio regulation), various sources of syngas can be applied to biosynthesize a large variety of chemicals.

## Supporting information

**S1 Fig. Enrichment of selected categories of GO biological processes for genes differential expression in fructose (R2YG), 100% CO (R2YC) or 50% CO (R2YH) induced fermentation compared to cellular growth stage (R2K).**
(TIF)

**S1 Appendix. List of GO biological processes for genes downregulated in pure CO fermentation compared to cellular growth stage.**
(XLTX)

**S2 Appendix. KEGG pathway enrichment analysis for genes significantly upregulated in 50% CO (R2YH) induced fermentation compared to cellular growth stage (R2K).**
(XLTX)

**S3 Appendix. Amino acid content comparative analysis in 100% CO and 50% CO fermentation.**
(XLSX)

**S4 Appendix. List of primers used in this study.**
(XLSX)

## Author Contributions

**Conceptualization:** Cheng Yanling, Wanqing Wang.

**Data curation:** Manyu Tang, Guoqiang Zhao, Shuang Wu, Wei Hua, Jingwen Qiang, Wanqing Wang.

**Formal analysis:** Manyu Tang, Xin Zhen, Guoqiang Zhao.

**Funding acquisition:** Wanqing Wang.

**Investigation:** Manyu Tang, Guoqiang Zhao, Wanqing Wang.

**Methodology:** Manyu Tang, Shuang Wu, Wei Hua.

**Project administration:** Wanqing Wang.

**Resources:** Xin Zhen.

**Software:** Xin Zhen, Guoqiang Zhao, Jingwen Qiang.

**Supervision:** Cheng Yanling, Wanqing Wang.

**Validation:** Manyu Tang.

**Writing – original draft:** Cheng Yanling, Wanqing Wang.

**Writing – review & editing:** Wanqing Wang.

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
