## [Decision Letter · Decision Letter 0]

22 Mar 2024

PONE-D-24-01154The metabolic pathways of carbon assimilation and polyhydroxyalkanoate production by Rhodospirillum rubrum in response to different atmospheric fermentationPLOS ONE

Dear Dr. wang,

Thank you for submitting your manuscript to PLOS ONE. After careful consideration, we feel that it has merit but does not fully meet PLOS ONE’s publication criteria as it currently stands. Therefore, we invite you to submit a revised version of the manuscript that addresses the points raised during the review process.

We look forward to receiving your revised manuscript.

Kind regards,

Shashi Kant Bhatia

Academic Editor

PLOS ONE

Journal Requirements:

Reviewers' comments:

Reviewer's Responses to Questions

**Comments to the Author**

1. Is the manuscript technically sound, and do the data support the conclusions?

Reviewer #1: Yes

Reviewer #2: Yes

Reviewer #3: Yes

2. Has the statistical analysis been performed appropriately and rigorously? 

Reviewer #1: Yes

Reviewer #2: Yes

Reviewer #3: Yes

3. Have the authors made all data underlying the findings in their manuscript fully available?

Reviewer #1: Yes

Reviewer #2: Yes

Reviewer #3: Yes

4. Is the manuscript presented in an intelligible fashion and written in standard English?

Reviewer #1: Yes

Reviewer #2: Yes

Reviewer #3: Yes

5. Review Comments to the Author

Reviewer #1: Major Comments:

Methods:

1. GC analysis of PHA, How did the authors analysed the monomeric composition of PHA in GC analysis, did they performed Mass spectroscopy or they used the standard of 3-HV. Explain and mention in the section?

2. Mention the name of target genes for the qRT-PCR analysis in the section.

3. Give a brief methodology (in Methods section) regarding the fed-batch fermentation of the R. rubrum in a 3-L bioreactor.

Results and discussion:

4. At 50% CO conditions, the bacterium uptake and utilization rates of CO and 3-HV accumulation were increased. Whereas, the genes which supply the precursor molecules such as propionyl-Co-A (genes in isoleucine degradation pathway in Fig. 6 i.e., Rru_A1946) are found to be less abundant in 50% CO condition. Explain and discuss.

5. Rewrite the sentence “transcriptomic data and RT-PCR expression assay revealed that the expression level of RuBisCO showed no variation which implied the CBB pathway was not essential for CO2 fixation under 50% CO gas atmosphere (Fig.5)”. The sentence sounds confusing, mention the substrate conditions used for the comparison.

6. Clearly discuss, whether R. rubrum adopt photoautotrophic mode or chemoautotrophic mode of nutrition in the syngas conditions (when hydrogen was added).

7. The fed batch process was performed with CO:H2 ratio at 3:1 and feeding the acetate. Why did author add 10 mM of acetate on 4th day of CO/H2 cultivation whereas acetate was added on 7th day of pure CO condition?

8. What is the monomeric composition of the PHA accumulated in CO:H2 ratio at 3:1 and 10 mM addition of acetate? Author have used the term PHA in this section and if the PHA monomer were analysed, author should discuss here.

9. Author should compare and discuss the fed batch results of R. rubrum using syngas to PHA production in the light of some relevant literature.

10. Conclusions: Concise the conclusion section, and shift the statements of results and discussion to relevant section.

Minor comments

11. Change the Rs. Rubrum to R. rubrum in the abstract and elsewhere.

12. What is RRNCO medium? Provide the details.

13. Use a consistent style of writing units, for example ΜL should be ml or mL, hour or h ?

14. This seems to be contradicting results, author should explain ?

15. Figure 1 and Fig. 7, the figure markers are not clear in the figures

16. Why did authors used different style of writing the heading in materials and methods section (e.g, first letter capital in “Monitoring of the Acetate Consumption” whereas some heading are in sentence case. Use a similar style.

17. Take care of spacing between the words throughout the manuscript.

18. Mention the source of the commercial kits used in the study, for example cDNA synthesis kit and library preparation kits etc.

19. Italicize the scientific name in the references and elsewhere.

Reviewer #2: The manuscript presents important new information on the influence of syngas composition on the monomeric distribution of PHBV in R. rubrum. However, I consider that the final text needs an in-depth review to be accepted by the journal. My comments are below:

- In the introduction section, there are more complete and current articles at shaken flasks or bioreactor scale that describe and analyze the influence of the CO fraction on the production of H2 and the accumulation of PHB with this microorganism (see Rodríguez et al., 2021. Biotechnology for Biofuels 14: 168 and Godoy et al 2024. Applied Microbiol. Biotechnol. 108(1), 258). These works must be included as references in the text.

- Materials and methods: in bacterial strain, growth and media subsection, authors must clear indicate that the microorganism was grown under photoheterotrophic conditions. Additionally, the characteristics of the UV lamp used in the light shaker should be included in this section.

- PHB extraction and quantification by GC analysis: Authors mention that a calibration curve has been used to quantify the intracellular content of the biopolymer. However, methanolysis with PHBV has a low yield (around 20% of methylesthers detected by GC/MS with respect to the initial amount treated in the calibration curve). How was the calibration curve made? Have the authors used PHB or PHBV? Has this low extraction yield been taken into account? Authors should clarify and expand on it in this section. Otherwise, NMR analysis is more accurate to determine the molar monomeric distribution of the biopolymer.

- Result section: In Fig. 1 authors explained that fructose was used as a sole carbon source in preliminarly cultures, but in text they wrote "the bacteria were firstly transferred to (...) medium containing glucose or CO as...". If they have feed fructose, it should be clear in the text. Moreover, in Fig 1.A, fructose profile must be included in order to facilitate the understanding of experimental results. In Fig 1C, leyend of condition 100%CO was the same as 20%CO. Authors should unify the format of this figure in the final version of the manuscript.

- Result section "fed-batch cultivation with R. rubrum on syngas". It is necessary a better explanation about how the experiment was carried out in the bioreactor and a brief description of the equipment. (for example, the temperature is not specified, nor the gas flow rate used, nor how the proportion of H2/CO fed is regulated). Moreover, in the text it is not clear the use of acetate, or why. On the other hand, I believe that these explanations should be included in the materials and methods section, where authors must mention the information included in Figure 8.A. Finally, what fraction of the outlet gas stream returns to the broth according to the scheme shown in Fig 8.A?

- Result section "fed-batch cultivation with R. rubrum on syngas": Why the authors started the fermentations with 2 g/l of initial biomass? Have they used the same RRNCO medium or the bioreactor runs were performed as a resting cell bioprocess? The evolution of biomass profiles in Fig. 8 suggests this idea, but authors have not clarify or mention in the text. This section must be highly improved. To enhance the understanding of the evolution of the experiments, the acetate concentration profiles in the runs carried out with pure CO and CO/H2 75/25 should be included in Figures 8E and 8F, respectively. Figure 8D does not provide any relevant information in this study. I suggest that the most powerful results obtained in the bioreactor runs must be included in a Table, in order to see the effect of H2 over biopolymer production and to compare with other studies available in the literature. On the other hand, why has the monomeric distribution of the PHA produced in the bioreactor was not been included in Fig 8, if it is the most important object of study in this article?

Finally, the quality of the figures in general must be improved. In the documentation available for revision, they appear very pixelated, so that it is difficult to read the axes of the figures and the text that appears in them.

Reviewer #3: The author studied the metabolic pathways of carbon assimilation and polyhydroxyalkanoate production by Rhodospirillum rubrum in response to different atmospheric fermentation, and it contain meaningful results and research findings. I would recommend to accept, with major revision as follow.

1. A comparison table should be provided that shows the growth of the strain and PHA productivity from Rhodospirillum rubrum using CO2 or CO, in comparison with the results of previous studies on PHA production.

2. It's necessary to investigate or provide opinions on whether the addition of hydrogen affects only CO2 conversion or if there are other aspects that it might influence within the PHA biosynthesis pathway.

3. There should be a detailed examination and comparison with previous research results on the PHA productivity per unit time and the conversion rate of the supplied carbon. Based on the results presented now, the PHA productivity appears to be very low compared to when other carbon sources were used, making it difficult to gauge the competitiveness in terms of cost-effectiveness.

6. PLOS authors have the option to publish the peer review history of their article (what does this mean?). If published, this will include your full peer review and any attached files.

Reviewer #1: No

Reviewer #2: **Yes: **Alberto Rodríguez

Reviewer #3: No

---

## [Author Response · Author response to Decision Letter 0]

8 May 2024

Dear Editor,

We are grateful for the opportunity to revise our manuscript entitled “The metabolic pathways of carbon assimilation and polyhydroxyalkanoate production by Rhodospirillum rubrum in response to different atmospheric fermentation” in light of the editorial and reviewer comments. We have provided a point-by-point response and uploaded the letter as a separate file labeled 'Response to Reviewers'.

---

## [Decision Letter · Decision Letter 1]

23 May 2024

PONE-D-24-01154R1The metabolic pathways of carbon assimilation and polyhydroxyalkanoate production by Rhodospirillum rubrum in response to different atmospheric fermentationPLOS ONE

Dear Dr. wang,

Thank you for submitting your manuscript to PLOS ONE. After careful consideration, we feel that it has merit but does not fully meet PLOS ONE’s publication criteria as it currently stands. Therefore, we invite you to submit a revised version of the manuscript that addresses the points raised during the review process.

We look forward to receiving your revised manuscript.

Kind regards,

Shashi Kant Bhatia

Academic Editor

PLOS ONE

Journal Requirements:

**Additional Editor Comments:**

The authors have addressed all the concerns in the revised manuscript. However, a few minor improvements in the revised manuscript can be incorporated, which are as:

1. The newly added table 4 can be presented in concise manner. I suggest to merge the column with similar informations and reference.

2. Abbreviate the polyhydroxybutyrate (PHB) and poly(3-hydroxybutyrate-co-3-hydroxyvalerate) (PHBV) should be mentioned at the place of their first appearance in the text.

3. Author can improve the resolution of images, for example, Figure 3A, description on x-axis are not clearly

visible.

4. Thoroughly check the manuscript for typographical errors (Example, R.rubrum mentioned in abstract and keywords, put a space in between R. and rubrum, it should be R. rubrum).

5. Italicise the Rhodospirillum rubrum in the legends of table 3 and 4 and elsewhere (if any).

6. Make a uniformity in writing the units as per the style of journal (example, g L-1 or g/L in table 3, table 4 and in the text elsewhere. Another example is, author are using ml for millilitre and L for litre).

Reviewers' comments:

Reviewer's Responses to Questions

**Comments to the Author**

1. If the authors have adequately addressed your comments raised in a previous round of review and you feel that this manuscript is now acceptable for publication, you may indicate that here to bypass the “Comments to the Author” section, enter your conflict of interest statement in the “Confidential to Editor” section, and submit your "Accept" recommendation.

Reviewer #1: All comments have been addressed

Reviewer #2: All comments have been addressed

Reviewer #3: All comments have been addressed

2. Is the manuscript technically sound, and do the data support the conclusions?

Reviewer #1: Yes

Reviewer #2: Yes

Reviewer #3: Yes

3. Has the statistical analysis been performed appropriately and rigorously? 

Reviewer #1: N/A

Reviewer #2: Yes

Reviewer #3: I Don't Know

4. Have the authors made all data underlying the findings in their manuscript fully available?

Reviewer #1: Yes

Reviewer #2: Yes

Reviewer #3: Yes

5. Is the manuscript presented in an intelligible fashion and written in standard English?

Reviewer #1: No

Reviewer #2: Yes

Reviewer #3: Yes

6. Review Comments to the Author

Reviewer #1: The authors have addressed all the concerns in the revised manuscript. However, a few minor improvements in the revised manuscript can be incorporated, which are as:

1. The newly added table 4 can be presented in concise manner. I suggest to merge the column with similar informations and reference.

2. Abbreviate the polyhydroxybutyrate (PHB) and poly(3-hydroxybutyrate-co-3-hydroxyvalerate) (PHBV) should be mentioned at the place of their first appearance in the text.

3. Author can improve the resolution of images, for example, Figure 3A, description on x-axis are not clearly

visible.

4. Thoroughly check the manuscript for typographical errors (Example, R.rubrum mentioned in abstract and keywords, put a space in between R. and rubrum, it should be R. rubrum).

5. Italicise the Rhodospirillum rubrum in the legends of table 3 and 4 and elsewhere (if any).

6. Make a uniformity in writing the units as per the style of journal (example, g L-1 or g/L in table 3, table 4 and in the text elsewhere. Another example is, author are using ml for millilitre and L for litre).

Reviewer #2: Thank you for introduce the changes suggested during the previous revision of the manuscript. In the current form, the article has much better quality and can be published.

Reviewer #3: Upon reviewing the revised manuscript, it is determined that the paper has been modified according to the reviewer's suggestions. The revised manuscript is considered acceptable for publication.

7. PLOS authors have the option to publish the peer review history of their article (what does this mean?). If published, this will include your full peer review and any attached files.

Reviewer #1: **Yes: **Vijay Kumar

Reviewer #2: No

Reviewer #3: No

---

## [Author Response · Author response to Decision Letter 1]

10 Jun 2024

Academic editor’s comments

1. The newly added table 4 can be presented in concise manner. I suggest to merge the column with similar informations and reference.

Response:

We feel great thanks for your professional review work on our article. According to your nice suggestion, we have merged the column with similar information (growth conditions, cultivation modes) and reference in table 4. It does look more concise now.

2. Abbreviate the polyhydroxybutyrate (PHB) and poly(3-hydroxybutyrate-co-3-hydroxyvalerate) (PHBV) should be mentioned at the place of their first appearance in the text.

Response:

Thanks for your careful checks. We have added the abbreviations of the polyhydroxybutyrate (PHB) and poly(3-hydroxybutyrate-co-3-hydroxyvalerate) (PHBV) at their first appearance in the text.

3. Author can improve the resolution of images, for example, Figure 3A, description on x-axis are not clearly visible.

Response:

Thank you for your suggestion and we have provided high quality figures in the final version of the manuscript to ensure clear visibility of the axes of the figures and the text that appears in them. But，we are sorry that during the image upload process, the system seems to automatically compress the image quality. We have resubmitted the figure 3. 

4. Thoroughly check the manuscript for typographical errors (Example, R.rubrum mentioned in abstract and keywords, put a space in between R. and rubrum, it should be R. rubrum).

Response:

We sincerely thank you for careful reading. As your suggestions, we have thoroughly checked the manuscript for typographical errors and corrected the R.rubrum into R. rubrum in the abstract, keywords and elsewhere. 

5. Italicise the Rhodospirillum rubrum in the legends of table 3 and 4 and elsewhere (if any).

Response:

We feel sorry for our carelessness. In our resubmitted manuscript, we have italicised the Rhodospirillum rubrum in the legends of table 3 and 4 and elsewhere.

6. Make a uniformity in writing the units as per the style of journal (example, g L-1 or g/L in table 3, table 4 and in the text elsewhere. Another example is, author are using ml for millilitre and L for litre).

Response:

Thank you for your kind suggestion. We have reviewed the whole paper and used a consistent style to rewrite the units. (for example, changing g/L into g l-1 in table3, 4 and anywhere, using l for litre)

---

## [Editor Report · Decision Letter 2]

13 Jun 2024

The metabolic pathways of carbon assimilation and polyhydroxyalkanoate production by Rhodospirillum rubrum in response to different atmospheric fermentation

PONE-D-24-01154R2

Dear Dr. wang,

We’re pleased to inform you that your manuscript has been judged scientifically suitable for publication and will be formally accepted for publication once it meets all outstanding technical requirements.

Kind regards,

Shashi Kant Bhatia

Academic Editor

PLOS ONE

Additional Editor Comments (optional):

The authors have revised the manuscript as suggested by reviewers, and now it can be accepted as it is.
---

## [Editor Report · Acceptance letter]

18 Jun 2024

PONE-D-24-01154R2 

PLOS ONE

Dear Dr. Wang, 

I'm pleased to inform you that your manuscript has been deemed suitable for publication in PLOS ONE. Congratulations! Your manuscript is now being handed over to our production team.

Kind regards, 

on behalf of

Dr. Shashi Kant Bhatia 

Academic Editor

PLOS ONE